# A Novel Vaccine Strategy to Prevent Cytauxzoonosis in Domestic Cats

**DOI:** 10.3390/vaccines11030573

**Published:** 2023-03-02

**Authors:** Pabasara Weerarathne, Rebekah Maker, Chaoqun Huang, Brianne Taylor, Shannon R. Cowan, Julia Hyatt, Miruthula Tamil Selvan, Shoroq Shatnawi, Jennifer E. Thomas, James H. Meinkoth, Ruth Scimeca, Adam Birkenheuer, Lin Liu, Mason V. Reichard, Craig A. Miller

**Affiliations:** 1Department of Veterinary Pathobiology, College of Veterinary Medicine, Oklahoma State University, Stillwater, OK 74078, USA; 2Department of Physiological Sciences, College of Veterinary Medicine, Oklahoma State University, Stillwater, OK 74078, USA; 3Oklahoma Animal Disease Diagnostic Laboratory, College of Veterinary Medicine, Oklahoma State University, Stillwater, OK 74078, USA; 4Department of Clinical Sciences, College of Veterinary Medicine, Oklahoma State University, Stillwater, OK 74078, USA; 5Department of Clinical Sciences, North Carolina State University, Raleigh, NC 27606, USA

**Keywords:** cytauxzoonosis, *Cytauxzoon felis*, adenoviral vector vaccines, domestic cats, tick-borne diseases, c88, cf76

## Abstract

Cytauxzoonosis is caused by *Cytauxzoon felis* (*C. felis*), a tick-borne parasite that causes severe disease in domestic cats in the United States. Currently, there is no vaccine to prevent this fatal disease, as traditional vaccine development strategies have been limited by the inability to culture this parasite in vitro. Here, we used a replication-defective human adenoviral vector (AdHu5) to deliver *C. felis*-specific immunogenic antigens and induce a cell-mediated and humoral immune response in cats. Cats (*n* = 6 per group) received either the vaccine or placebo in two doses, 4 weeks apart, followed by experimental challenge with *C. felis* at 5 weeks post-second dose. While the vaccine induced significant cell-mediated and humoral immune responses in immunized cats, it did not ultimately prevent infection with *C. felis*. However, immunization significantly delayed the onset of clinical signs and reduced febrility during *C. felis* infection. This AdHu5 vaccine platform shows promising results as a vaccination strategy against cytauxzoonosis.

## 1. Introduction

*Cytauxzoon felis* is an apicomplexan parasite that is related to *Theileria* spp., *Babesia* spp., and *Plasmodium* spp. [1,2,3]. While *C. felis* is the only known species of the genus in the United States (US), other *Cytauxzoon* spp. have been identified in several countries from five different continents [4,5,6,7,8,9,10,11,12]. Infection of domestic cats with *C. felis* causes cytauxzoonosis, a fatal tick-borne disease [13], and cases of cytauxzoonosis have been documented all over the southcentral, southeastern, and mid-Atlantic US [2,13,14]. Following infection with *C. felis*, domestic cats experience severe disease progression characterized by fever, lethargy, inappetence, lethargy, depression, dehydration, dyspnea, hemolytic crisis, and icterus. Mortality can be high without treatment and supportive care [2,15]. Currently, there is no definitive cure for cytauxzoonosis, highlighting the critical need for development of therapies and vaccines to protect domestic cats from this highly fatal disease.

The current standard of care for cytauxzoonosis is a combination therapy with atovaquone and azithromycin [2,16]. However, this therapy only has a moderate success rate, and only if treatments are initiated early in the course of infection [16]. The only preventive measures currently available are acaricides [17,18] administered to cats to prevent tick attachment or limit tick feeding and interrupting *C. felis* transmission from infected ticks. However, this approach may be cost-prohibitive or difficult to apply routinely to cats in enzootic areas. Development of an effective vaccine against cytauxzoonosis would provide a key prevention strategy in an integrated control program for limiting cytauxzoonosis in domestic cats. In 2013, Tarigo et al. [14] sequenced and annotated the *C. felis* genome to identify immunogenic vaccine candidates and found protein cf76; that resembled p67, a vaccine candidate for bovine theileriosis [19]. The protein sequence of cf76 was found to be conserved between samples collected from different geographic regions and was shown to be an immunogenic antigen expressed in the schizogenous life stage of *C. felis* [14]. In 2018, Schreeg et al. [20] screened 673 putative *C. felis* proteins for antigenicity using protein microarray, including cf76. By comparing these proteins in sera from *C. felis* naïve and infected cats, 33 immunogenic antigens were identified and incorporated into two different DNA-expression library vaccines. One of the vaccines comprised of 32 of the 33 antigens, whereas the other vaccine contained the carboxyl terminus and the full length of cf76. The vaccines were given in three doses followed by *C. felis* challenge. Unfortunately, neither of these vaccines prevented cats from *C. felis* infection or reduced clinical disease following exposure. However, these findings identified other immunogenic proteins corresponding to *C. felis* transmembrane antigens, such as protein expressed from contig00088:95434-96586(−) (hereafter termed “c88”), that also had high seroreactivity to infected feline serum, indicating a highly immunogenic potential for use in vaccine development.

Although development of vaccines against parasitic diseases has been fraught with challenges, viral vector vaccines have shown promising results against intracellular parasites [21,22,23,24,25,26]. Adenoviruses were previously recognized as an efficient vector for gene therapy due to their increased infectious properties [27,28,29,30], and in recent years, adenoviruses have been widely used as vaccine vectors since the virus genome can be easily manipulated to deliver an antigen of interest and increase the efficacy of the vaccine through intracellular production of vaccine antigen [31,32,33]. As *C. felis* is an intracellular parasite, a cell-mediated immune response is likely crucial to eliminate infected cells and reduce the parasite growth. A vaccine that could deliver and express vaccine antigen in a way that would mimic natural infection would be favorable to initiate a stronger humoral and cellular immune response. In the current study, we employed a replication-defective human adenovirus serotype 5 (AdHu5) vector vaccine, encoding two putative proteins of *C. felis*, c88 and cf76, in attempt to prevent cytauxzoonosis in domestic cats. We hypothesized that intracellular expression of *C. felis*-specific proteins would induce a pronounced cell-mediated and humoral immune response that would protect from *C. felis* infection and thereby reduce clinical signs and disease severity of acute cytauxzoonosis (Figure 1). 

## 2. Materials and Methods

### 2.1. Construction of Ad-C88/CF76 Overexpression Vector and Generation of Adenovirus

Development and production protocols of AdHu5-C88 and AdHu5-CF76 vectors were adapted and performed according to previously published methods [34,35,36]. Briefly, C88/CF76 cDNA was PCR-amplified from a C88/CF76 HIS-vector (GenScript Biotech, Piscataway, NJ, USA) with Q5^®^ High-Fidelity DNA Polymerase (New England Biolabs, Ipswich, MA, USA) and proprietary primers. The PCR product was then inserted into pENTR vector through Sac I-EcoR I sites to express c88 and cf76 proteins. The construction of a full-length DNA clone encoding the recombinant adenovirus was performed by transferring the c88 or cf76 gene from pENTR vector to pAD-DEST vector via LR recombination reaction (Invitrogen, Carlsbad, CA, USA, Catalog Numbers 12535-019 and 12535-027). The clone was confirmed by sequencing, and generation of recombinant adenoviruses was performed by digesting 5 µg of adenoviral vector by PAC I restriction enzyme (New England Biolabs, Ipswich, MA, USA), followed by transfection of 1 µg of digested vector into HEK293A cells in Dulbecco’s Modified Eagle Medium (DMEM) supplemented with 10% fetal bovine serum and 0.1% penicillin and streptomycin (ThermoFisher Scientific, Carlsbad, CA, USA) per manufacturer’s instructions. After 48 h, cells were transferred into 10 cm plates and cultured for additional 14 days. Viral amplification was achieved by infecting HEK293A cells with crude c88 and cf76 adenoviruses into HEK293A cells in 10-cm plates. Adenoviruses were purified by using Adeno-X™ Maxi Purification Kit (Invitrogen, Waltham, MA, USA). Virus titers were determined by infecting HEK293T cells with a series dilution of purified adenovirus for 48 h and by using the Adeno-X™ Rapid Titer Kit (Clontech, Mountain View, CA, USA). The titers of Ad-c88 and Ad cf76 were 3.4 × 10^9^ and 1.9 × 10^9^ pfu/mL, respectively.

### 2.2. Animals

A total of twelve (*n* = 12) spayed/neutered specific pathogen free (SPF) domestic cats were obtained from the Andrea D. Lauerman Specific Pathogen Free Feline Research Colony at Colorado State University (Fort Collins, CO, USA). Cats intended for the vaccine (*n* = 6) and sham-vaccine (*n* = 6) were group housed within an AAALAC International-accredited animal facility at Oklahoma State University (Stillwater, OK, USA) and were fed dry/wet food with access to water ad libitum for the duration of the study. All the cats were examined clinically for their health status and were allowed to acclimate for 21 days prior to initiation of the study. Temperature sensing microchips (Bio Medic Data Systems, Seaford, DE, USA) were subcutaneously implanted in the dorsum, and body weights, temperatures (acquired via thermal microchips), and blood samples were obtained prior to vaccination on day 0 to serve as a baseline measurement. All cats were considered in good health and confirmed to be *C. felis*-negative by ddPCR at onset of the study [37]. 

### 2.3. Immunization and Challenge of Cats

The study was conducted over a 12-week period (Figure 2). Adenovirus vectors containing the c88 gene (7.14 × 10^8^ IFU per dose) and cf76 gene (7.14 × 10^8^ IFU per dose) were combined (0.21 mL of each vector, 0.42 mL total per dose) and administered intramuscularly (IM) to a subset of cats (*n* = 6) in two doses, one at week 0 and again at week 4. The control cats (*n* = 6) received 0.42 mL of phosphate buffered saline (PBS) (Fisher Scientific, Pittsburgh, PA, USA). Following each immunization, cats were housed in individual kennels for 72 h for observation. 

At 9 weeks from the first dose of the vaccine (0 dpi), all cats (*n* = 12) were challenged with *C. felis* infection using established transmission methods [38,39] and were housed in individual kennels until the end of the study. Briefly, laboratory-reared nymphs of *Amblyomma americanum* were acquisition fed on a donor cat with chronic, subclinical *C. felis* infection. Engorged nymphs were collected, stored in a humidity chamber, and allowed to molt to adults. Adult *A. americanum* (*n* = 50/cat) were then transmission fed on all vaccinated (*n* = 6) and sham-inoculated control (*n* = 6) cats until female ticks had fed to repletion (approximately 12 days). Principal cats were monitored daily for clinical signs of cytauxzoonosis, as described below in Section 2.9.

### 2.4. Sampling

Blood samples (up to 4 mL per timepoint) were collected from all the cats via jugular venipuncture at the time points indicated in (Figure 2). Blood samples were collected on 0, 7, 12, 15 and 21 days post infestation (dpi). Cats were lightly sedated for blood collection, tick infestation and tick removal procedures as previously described [40]. Collected blood samples were used for complete blood counts (CBC), flow cytometry, to evaluate cellular and humoral immune responses, and to quantify parasite load after tick infestation as outlined below. At the end of the study, vaccinated cats were humanely euthanized (pentobarbital >80 mg/kg) and necropsied to collect tissue samples. Sham-vaccinated cats (*n* = 2) that developed severe cytauxzoonosis were also euthanized according to the same protocol. Remaining sham-vaccinated cats (*n* = 4) were treated with atovaquone (15 mg/kg PO q8h) and azithromycin (10 mg/kg PO q24h).

### 2.5. Flow Cytometry

Flow cytometry analysis of circulating lymphocyte lineages was performed as previously described [41]. Briefly, 50 μL of EDTA-treated blood was treated with mouse monoclonal antibodies targeting CD4, CD8 and CD21 feline cell markers (Fisher, clone 3-4F4, FITC; Southern Biotech, clone fCD8, PE; and Bio-Rad, CA2.1D6, AF647 respectively) to determine the percentage of each cell type. Unstained and single stained samples were used as controls. BD FACSAria™ SORP instrument (Becton Dickinson, San Jose, CA, USA) containing BD FACSDiva™ Software (Diva 9.0.1., San Jose, CA, USA) was used to obtain data, and were analyzed using FlowJo 10.8.0. (Ashland, OR, USA).

### 2.6. ELISpot Assay

c88- and cf76- specific cytotoxic T-cell recall response was determined using the cat IFN-γ ELISpotPLUS kit (Mabtech AB, Nacka Strand, Sweden) according to manufacturer’s instructions. Briefly, PBMCs were isolated from the blood samples collected from each cat using density gradient at each sample collection point (Figure 2). Cells were then stimulated with 5μg/mL of c88 or cf76 recombinant protein (GenScript Biotech, Piscataway, NJ, USA) and added (3 × 10^5^ cells/well in 100 μL) to the anti-IFN-γ monoclonal antibody (mAb MT131, Mabtech AB) precoated 96-well plate in duplicate. Cells stimulated with PHA (4 μg/mL) were used as a positive control and unstimulated cells were used as a negative control for each cat. The plate was incubated for 48 h at 37 °C with 5% CO_2_. Following the incubation, cells were washed with PBS, and biotinylated detection antibody (mAB MT114, Mabtech AB) was added and incubated for 2 h at room temperature. Next, unbound detection antibodies were washed, streptavidin-ALP was added and further incubated for 1 h. After washing with PBS, spots were developed on the membrane by adding the substrate solution (BCIP/NBT-plus) provided with the kit. Plates were extensively washed in running tap water and allowed to air dry overnight before enumerating the spot-forming cells using an Olympus SZX7 microscope (Olympus, Shinjuku City, Tokyo, Japan). Spot-forming cell numbers in the negative control were subtracted from the numbers in protein-stimulated cells to get the actual number of T-cells responded to each protein.

### 2.7. IgM and IgG ELISAs

Indirect ELISAs were performed to detect anti-c88 and anti-cf76 IgM and IgG antibodies in response to vaccination and *C. felis* infection using plasma separated from EDTA-treated blood samples. Anti-c88 specific IgM and IgG levels were measured using a previously published assay [42]. Briefly, 96-well Microplates (Corning Inc. Glendale, AZ, USA) were coated with c88 protein (5 μg/mL for IgM, 2.5 μg/mL for IgG; 100 μL/well) diluted in carbonate buffer (4 °C, overnight). Blocking was done by adding 2% bovine serum albumin (Promega, Madison, WI, USA) in TEN buffer (200 μL/well) (4 h at room temperature). Plasma samples were diluted at 1:500 in ELISA diluent, added to respective wells (100 μL/well) in duplicates and incubated for 1 h at 37 °C. Plasma from a naturally infected cat with acute cytauxzoonosis and a *C. felis* PCR negative SPF cat was used as the positive and negative controls respectively. ELISA diluent alone served as the blank. Following the incubation, plates were washed five times with 0.2% /tween in TEN buffer. Diluted HRP conjugated IgM (1:80,000) or IgG (1:60,000) antibodies (Bethyl, Montgomery, TX, USA) were added to the wells (100 μL/well) and further incubated for 1 h at 37 °C. Next, plates were washed and 100 μL of 1-Step™ Ultra TMB-ELISA Substrate Solution (Thermo Fisher Scientific, Waltham, MA, USA) was added (room temperature, 10 min). Reaction was discontinued by adding 75 μL of ELISA Stop Solution (Thermo Fisher Scientific, Waltham, MA, USA). Absorbance was measured at 450 nm using Cytation™ 5 multimode reader (BioTek, Winooski, VT, USA). Data were presented as percent positives. 

To evaluate the production of anti-cf76 IgM and IgG in response to vaccination and *C. felis* infection, an ELISA was performed similar to the aforementioned c88 assay, with a few modifications. Plasma samples were diluted 1:500 in ELISA diluent containing *E. coli* lysate (0.2 μg/mL) and incubated for 3 h on a plate shaker at 450 rpm before adding to the blocked plate (to reduce background). Detection antibodies were added at 1:40,000 for both IgM and IgG. Results were recorded as the average absorbance values. Positive cut-off values for each antibody type were set by calculating the value of the negative control + 3 standard deviations as previously described [43].

### 2.8. Droplet Digital PCR (ddPCR)

Parasite load in the blood samples were quantified using a probe based ddPCR assay as previously described [37]. In brief, DNA was extracted from 100 μL of EDTA-treated blood. Reaction mixes were generated by mixing 13.3 μL of ddPCR master mix [37] with 8.8 μL of DNA extraction. Next, 20 μL were transferred to respective wells of DG8™ Cartridges (Bio-Rad Laboratories, Inc. Hercules, CA, USA) followed by 70 μL of Droplet Generation Oil for Probes (no dUTP) (Bio-Rad Laboratories, Inc. Hercules, CA, USA). Aqueous droplets in oil emulsions were generated in QX200™ Droplet Generator (Bio-Rad Laboratories, Inc. Hercules, CA, USA). Samples were added (40 μL) to respective wells in a 96-well plate, sealed and PCR was carried out in C1000 Touch™ Thermal Cycler (Bio-Rad Laboratories, Inc. Hercules, CA, USA) [37]. Using a QX200™ Droplet Reader (Bio-Rad Laboratories, Inc. Hercules, CA, USA), droplets were then analyzed for absolute quantification of the target DNA. All samples were run in duplicate, with a (i) positive control, (ii) negative control, and (iii) no template control. Thresholds were set and absolute copy numbers were calculated as previously described [37].

### 2.9. Clinical Scoring

Cats were monitored by a licensed veterinary practitioner at least once a day for any discomfort and morbidity for the duration of the study. Body temperatures were recorded daily, and body weights were determined weekly. Following tick infestation (week 9), body temperatures were determined three times a day. In addition, a quantitative clinical scoring system was developed and performed following the tick infestation, and evaluators were blinded to study groups to ensure an unbiased assessment and determination of treatments needed. Clinical parameters (Table 1) included rectal temperature, mucous membrane capillary refill time (MM-CRT), activity, appetite, respiratory effort, icterus, dehydration, and pain. Each factor was assigned a score of 0 (healthy), 1 (mild), 2 (moderate), or 3 (marked), and summated clinical scores for each cat were calculated daily by adding the values of each clinical parameter. Total score of 0–3 was considered healthy, 4–6 as mild development of cytauxzoonosis, 7–10 as moderate development of cytauxzoonosis, and scores > 11 indicated severe cytauxzoonosis. Elevated body temperature above 104 °F combined with marked lethargy and/or anorexia prompted initiation of atovaquone and azithromycin treatments. Cats with more than 5% dehydration received fluids subcutaneously, and cats with pain scores above 3 received buprenorphine at the labelled dose (Simbadol, Zoetis, Florham Park, NJ, USA). Cats that developed a clinical score > 12 were humanely euthanized. Cats receiving treatment but remaining at a summated clinical score > 10 with no improvement after 48 h of treatment were humanely euthanized. Prior to euthanasia, study animals were anesthetized by intramuscular injection of ketamine (4 mg/kg), dexmedetomidine (0.02 mg/kg), and butorphanol (0.4 mg/kg), and then humanely euthanized with pentobarbital overdose (>80 mg/kg) and necropsied to collect tissue samples.

### 2.10. Histopathology

Histopathology was performed as previously described [41]. In brief, two sham-vaccinated cats were humanely euthanized during this study due to severe cytauxzoonosis and were necropsied at 17 and 18 dpi, respectively. The remaining sham-vaccinated cats (*n* = 4) survived *C. felis* infection and were adopted out per OSU policy. One vaccinated cat was humanely euthanized due to severe cytauxzoonosis and necropsied at 18 dpi (*n* = 1). The remaining vaccinated cats (*n* = 5) survived *C. felis* challenge and were humanely euthanized and necropsied at 23 dpi per NIH Guidelines for Research Involving Recombinant or Synthetic Nucleic Acid Molecules [44]. Heart, lung, liver, kidney, spleen, skeletal muscle, popliteal lymph node, and brain tissues were collected, halved, and frozen at −80 °C or fixed in 10% neutral-buffered formalin for 7 days prior transferring to 70% ethanol. Paraffin embedded tissue sections were trimmed to 5 µm, collected on positively charged slides, and then stained with hematoxylin and eosin (H&E) for microscopic evaluation. A histological score of increasing severity (0–4) was assigned to all tissue samples based on previous studies [40,45,46] and the following criteria: degree of inflammatory infiltration and/or tissue damage, degree of vascular occlusion, and quantitation of intralesional schizonts. A board-certified veterinary pathologist blinded to study groups evaluated and scored the tissue samples to ensure reproducibility and scientific rigor.

### 2.11. Quantification of Large Granular Lymphocytes

Blood smears were prepared and evaluated for large granular lymphocytes (LGLs) by a board-certified clinical pathologist blinded to study groups. Blood films were stained with methanolic Romanowsky stain to better visualize the presence of cytoplasmic granules in cells such as large granular lymphocytes. Three hundred cell differential white cell counts were done from each sample to increase the precision of the differential counts. Large granular lymphocytes, if present, were counted separately from other lymphocytes. To classify cells as LGLs, the cell had to have increased amounts of cytoplasm (cytoplasm visible around at least half of the nuclear perimeter) with 2 or more well-defined purple cytoplasmic granules.

### 2.12. Statistical Analysis

All cats in this study were assigned randomly to vaccinated and sham-vaccinated groups. All ddPCR, ELISpot assay and ELISAs were performed in duplicates. Results were statistically analyzed using Graph Pad Prism 9.0 Software (La Jolla, CA, USA), and when applicable, presented as mean ± SEM. To compare differences between vaccination groups over time, repeated measures ANOVA with mixed effects analysis were used. *p*-values < 0.05 were considered statistically significant.

## 3. Results

### 3.1. Pronounced Cell-Mediated Immune Response to Vaccination

ELISpot assay was performed to evaluate the cytotoxic T-cell (CTL) response to the *C. felis*-specific proteins, c88 and cf76. Following the first dose of the vaccine on week 0, a mild cell-mediated response to both c88 and cf76 vaccine epitopes was detected at weeks 2 and 4 in vaccinated cats (Figure 3). By week 6, (two weeks post-boost) the cellular response was significantly increased for both c88 (*p* < 0.0143) and cf76 (*p* < 0.0262) vaccine epitopes in the vaccinated group compared to the sham-vaccinated group. The CTL response in the vaccinated group decreased slightly at week 9 (0 dpi), but was still markedly elevated against the cf76 epitope (*p* < 0.0177) and the c88 epitope (*p* < 0.0565). While the cell-mediated immune response for both vaccine epitopes decreased during tick infestation and *C. felis* infection (7 and 12 dpi) in the vaccinated group, the response against c88 was still higher in vaccinated cats compared to unvaccinated cats at 7 dpi (*p* < 0.422). At 15 dpi, there was a slightly increase in the cellular response to both the vaccine epitopes in unvaccinated cats, but this was not significant.

### 3.2. Increased Anti-c88 IgG Levels in Immunized Cats 

IgG and IgM antibodies against *C. felis*-specific proteins c88 and cf76 were evaluated in plasma from all animals over the course of the study using an indirect ELISA (Figure 4). Anti-c88 IgG antibodies were significantly elevated in vaccinated cats beginning at week 4 (*p* < 0.0001) and remained elevated throughout the entire course of the study (Figure 4a). Following the *C. felis* challenge (week 9, 0 dpi), anti-c88 IgG antibodies increased further in vaccinated cats and were significantly elevated over pre-challenge (0 dpi) levels at both 15 and 21 dpi (*p* < 0.0001). Anti-c88 IgG antibodies in vaccinated cats were also significantly higher during *C. felis* challenge than sham-vaccinated cats (*p* < 0.0001). Anti-c88 IgG levels in sham-vaccinated cats remained undetectable throughout the study until 12 dpi, when IgG levels increased substantially, albeit to a lesser degree than observed in vaccinated cats. 

Unlike anti-c88 IgG, a significant difference in anti-c88 IgM levels were not observed between groups in response to immunization (Figure 4b). However, following tick infestation and *C. felis* challenge, IgM levels increased in both vaccinated and sham-vaccinated animals by 12 dpi, and was significantly elevated in all cats by 21 dpi (*p* < 0.0001). Interestingly, IgM levels observed in the sham-vaccinated group was significantly higher than the vaccinated group at day 15 and 21 post-infection (*p* = 0.0004 and *p* < 0.0001, respectively).

Interestingly, anti-cf76 IgG and IgM antibodies were not detected in vaccinated cats prior to *C. felis* challenge. An increase in anti-cf76 IgG was observed at 21 dpi in both vaccinated and sham-vaccinated cats (Figure 4c), however, this was only significant in sham-vaccinated cats (*p* < 0.0001). Anti-cf76 IgM antibodies increased earlier than IgG and were detectable by 12 dpi in vaccinated cats and at 15 dpi in sham-vaccinated cats (Figure 4d). Anti-cf76 IgM antibodies were significantly elevated at 21 dpi (*p* < 0.0001) in both study groups, and vaccinated cats tended to have higher IgM levels compared to the sham-vaccinated cats, although this was not statistically significant.

### 3.3. Changes in the CD4+ and CD21+ Cells following Vaccination 

Changes in the number of CD4+, CD8+, and CD21+ cells were analyzed by flow cytometry and plotted against time (Figure 5). There was no difference in the number of CD8+ cells between the vaccinated and sham-vaccinated group following the vaccine regimen (Figure 5a). However, in both groups, CD8+ cell counts were significantly elevated at 21 dpi, indicating a pronounced CTL response to *C. felis* challenge. Additionally, there was no significant difference in CD4+ cells between the two groups (Figure 5b) over the course of the study, but at 15 dpi, CD4+ were significantly decreased in both groups in response to *C. felis* challenge. During the immunization phase, CD21+ cells were slightly elevated in vaccinated cats at week 2 and week 6, suggesting a B-cell response to immunization following each dose (Figure 5c). However, this was not significant, and there was no significant difference in the number of CD21+ cells observed between the vaccinated and sham-vaccinated group over the course of the study. Interestingly, CD21+ cells decreased significantly following *C. felis* challenge and was most pronounced at 12, 15, and 21 dpi. No significant difference in the B-cell:T-cell ratio (Figure 5d) or CD4+:CD8+ ratio (Figure 5e) was observed between the two groups at any point during the study. However, there was a transient increase in the CD4+:CD8+ ratio in response to *C. felis* challenge at 12 dpi, followed by a significant decrease in both the B-cell:T-cell ratio and the CD4+:CD8+ ratio at 21 dpi.

### 3.4. Vaccination and C. felis Infection Increased the LGL Levels in Cats 

From each of the blood smears prepared, 300 white blood cells were screened per slide and number of LGLs was enumerated. Data were presented as the total number of LGLs/μL in each blood sample (Figure 6). During the immunization phase, there was a significant increase in LGLs in vaccinated cats following the booster dose at week 6 (*p* = 0.0004), and also at week 8 (*p* = 0.0466) compared to sham-vaccinated cats (Figure 6a). LGLs decreased thereafter and remained low until 21 dpi, when LGL counts increased significantly (*p* < 0.0001) in both groups in response to *C. felis* challenge (Figure 6b). No significant difference in LGL counts was observed between groups in response to *C. felis* challenge (0 dpi–21 dpi). 

### 3.5. Delayed Onset of Cytauxzoonosis and Decreased Febrility

Following tick infestation, cats were examined and evaluated daily (every 24 h) for different clinical parameters, as outlined in Table 1. Overall clinical scores for both the vaccinated and sham-vaccinated groups increased slightly at 3 dpi but remained low until 14 dpi (Figure 7a). Clinical scores of the sham-vaccinated cats increased earlier in the course of *C. felis* challenge, and were significantly elevated over vaccinated cats at 15 and 16 dpi (*p* = 0.0206 and 0.0424, respectively). A considerable delay in the onset of clinical signs was appreciable in vaccinated cats until they peaked at 18 dpi, and at 19 dpi, they were transiently elevated over sham-vaccinated cats that were recovering from acute infection (*p* = 0.0102). Vaccinated cats also exhibited lower average body temperatures during *C. felis* challenge when compared to sham-vaccinated cats (Figure 7b). Average body temperatures of vaccinated cats remained within ±1 °F of the normal reference range (98.1–102.1 °F) for the duration of *C. felis* challenge. In sham-vaccinated cats, average body temperatures increased significantly over at 13 dpi and remained elevated through 16 dpi. Four of the six sham-vaccinated cats developed temperatures over 104 °F (40 °C) at multiple timepoints, whereas the highest temperature reached in the vaccinated group was 103.1 °F (39.5 °C) in one cat.

### 3.6. Outcome of C. felis Challenge

Droplet digital PCR targeting the *C. felis* cox3 gene was conducted to quantify the parasite load in vaccinated and sham-vaccinated animals following immunization and tick-transmission challenge (Figure 8). ddPCR results were negative at week 0 and week 9 for all the cats, confirming that none of the study animals were exposed to *C. felis* infection prior to or during the course of the immunization phase. At 7 dpi, a low level of cox3 expression was detected in one of the cats in the vaccinated group (6 copies/µL). By 12 dpi, all cats had tested positive for *C. felis* and remained positive for the course of the study period. No significant difference in parasite load was observed over time between vaccinated and sham-vaccinated cats (interaction, *p* > 0.9999). 

During this study, *n* = 3 sham-vaccinated cats developed clinical signs of severe cytauxzoonosis at 15 dpi, which prompted initiation of atovaquone and azithromycin treatments (body temperature > 104 °F with marked lethargy and/or anorexia). Another *n* = 2 sham-vaccinated cats developed clinical scores greater than 12 at 18 and 19 dpi respectively, warranting humane euthanasia. Of the three sham-vaccinated cats that received atovaquone and azithromycin treatments, all three survived. One vaccinated cat developed a clinical score greater than 12 at 19 dpi and was humanely euthanatized. Three vaccinated cats also developed clinical signs of severe cytauxzoonosis that warranted atovaquone and azithromycin treatments, although later in the course of the *C. felis* challenge than in the sham-vaccinated cats (*n* = 1 at 16 dpi and *n* = 2 at 19 dpi). All three of these cats improved rapidly and treatment was discontinued at 21 dpi. Two vaccinated cats did not require therapy. In total, five of the six vaccinated cats survived *C. felis* challenge.

### 3.7. Pathology of C. felis Infection

As outlined above, three cats (one vaccinated and two sham-vaccinated) developed clinical scores sufficient to warrant euthanasia. At necropsy, the thoracic cavities of all animals contained 50–150 mL of thick yellow transudate, and the lungs were diffusely mottled dark red and exuded large amounts of foamy fluid on cut section. The spleens of all three animals were markedly enlarged, and the liver of one sham-vaccinated animal was moderately enlarged and diffusely firm with an enhanced reticular pattern. The other vaccinated cats that survived *C. felis* challenge (*n* = 5) were humanely euthanized and necropsied at 23 dpi per NIH Guidelines for Research Involving Recombinant or Synthetic Nucleic Acid Molecules [44]. The mucous membranes, sclera, adipose tissue, and abdominal viscera of these animals were minimally to mildly icteric, and the pericardial sac and thoracic cavity of these cats contained a small to moderate amount of straw-colored effusion. The lungs of these animals were moderately edematous and their spleens were moderately enlarged. 

Histologically, sham-vaccinated cats that did not survive *C. felis* challenge (*n* = 2) exhibited slightly increased perivascular and parenchymal inflammation in the heart (Figure 9a), mildly increased vascular occlusion and intravascular histiocytes with schizont formation in the lungs (Figure 9b) and spleen, and marginally increased vascular occlusion and parenchymal inflammation in the kidney (Figure 9c) when compared to the vaccinated cat that did not survive *C. felis* challenge (*n* = 1); although these were not statistically significant (Appendix A). However, total pathology scores in the kidney were significantly elevated in sham-vaccinated cats that did not survive *C. felis* challenge (*n* = 2) when compared to the vaccinated cat that did not survive *C. felis* challenge (*n* = 1, *p* = 0.0141) and vaccinated cats that survived *C. felis* challenge (*n* = 5, *p* = 0.0270). Additionally, there was a noticeable trend for an elevated total pathology score in the heart (*p* = 0.061) and spleen (*p* = 0.116) of sham-vaccinated cats that did not survive *C. felis* challenge (*n* = 2) when compared to the vaccinated cat (*p* = 0.0141) that did not survive *C. felis* challenge (*n* = 1). Significant pathology of the brain (*p* < 0.0001) was observed in both sham-vaccinated cats (*n* = 2) and the vaccinated cat (*n* = 1) that did not survive *C. felis* challenge when compared to vaccinated cats that survived *C. felis* challenge (*n* = 5); predominately characterized by perivascular and parenchymal inflammation with histiocytosis (Figure 9d).

## 4. Discussion

One of the hurdles in using adenoviral vectors in humans is the host response against these viruses that may neutralize their activity [21,29,47]. Previous studies have successfully used AdHu5 vectors to develop vaccines against feline diseases [36,48], and utilizing a human adenovirus in a feline system provides an added advantage of not being affected by pre-existing neutralizing antibodies and cytotoxic T-cells. The ultimate goal of employing a replication-defective AdHu5 vector vaccine in this study was to effectively deliver antigen-coding genes inside host immune cells that could then express the antigens and initiate a sufficient humoral and cellular immune response to *C. felis*-specific proteins, c88 and cf76. These two antigens were selected based on previous studies that showed that these proteins were expressed during the schizogenous phase of *C. felis* (the most crucial stage for parasite replication and clinical disease) and are known to be highly immunogenic, making them promising vaccine candidates [14,20,49,50]. We then demonstrated a significant cell-mediated (c88 and cf76) and humoral immune response (c88 only) to vaccine epitopes in vaccinated cats. Our results indicate that the expression of the intended antigens using the AdHu5 vector was fully successful for c88, and partially successful for cf76 (as only a transient cell-mediated immune response was observed). Immune cell and antibody responses were enhanced and peaked at week 6 following the second immunization, indicating the activation of acquired immune response. 

One of the benefits of using a replication-defective live viral vector is the opportunity to elicit a robust, longer lasting cellular response in addition to the humoral response. However, despite the promising humoral response, the cell-mediated immune response was not sustained in the vaccinated cats following the booster dose and decreased over time for both vaccine epitopes. A similar vaccine study by Svitek et al. [25] assessed the schizont-specific CD8+ central memory T-cells against *Theileria parva* and also showed an increase in the cellular immune response following the booster dose compared to the prime which eventually decreased overtime. However, in the present study, following the infection with *C. felis*, an increase in the CTL response was not observed for the vaccine epitopes in the vaccinated cats as expected until 15 dpi. A similar trend was also observed in the sham-vaccinated cats following the infection (Figure 3). Though there is a paucity of information on pathogenicity of *C. felis* and immune response to *C. felis* infection, we hypothesize that this could be due to T-cell impairment during *C. felis* infection. This was further supported by the decrease of T-cell counts following the infection (Figure 5). As *C. felis* is an intracellular parasite that initially infects and replicates in macrophages of its host, it should possess defensive mechanisms to dampen and evade the host immune responses in order to sustain replication inside host cells [51]. The delayed cell-mediated immune response seen in the present study following the infection could be due to the inhibition of cytokine expression and antigen presentation of macrophages [52]. As such, recruitment and activity of cytotoxic T-cells may have been delayed following the infection. Though the cell-mediated immune response was recovered by 15 dpi, the response observed in the vaccinated cats was much lower than what was observed in the sham-vaccinated cats. One possible explanation may be that the high load of vaccine epitopes presented during the immunization process may have resulted in T-cell exhaustion in vaccinated cats during *C. felis* infection [53], although our flow data showed no data to support this. Regardless, it is presumable that a sustained CTL response may have been more effective at neutralizing intracellular *C. felis* infection, and future studies aimed at producing a more sustained CTL response will likely prove more efficacious. 

Similar to data published by Schreeg et al. [20], the c88 antigen exhibited very high immunogenicity in the present study. Anti-c88 IgG levels were significantly elevated in vaccinated cats throughout the study, which were further elevated following infection with *C. felis* (Figure 4a). However, in contrast to the findings of the previous studies [14,20], cf76 did not exhibit high immunogenic characteristics in the present study. Vaccination did not induce anti-cf76 IgG or IgM production (Figure 4c,d), although a pronounced CTL response was observed in immunized cats against this epitope (Figure 3b). The timing and magnitude of the antibody response observed against cf76 after 15 days (for IgM) and 21 days (for IgG) of infection in the sham-vaccinated group infers that cf76 is not as immunogenic as c88. As such, cf76 may not be a suitable vaccine candidate or a molecular target for early detection of cytauxzoonosis in downstream studies.

Though a significant difference in the B- and T-cell populations was not observed between the vaccinated and sham-vaccinated groups, a slight increase in CD21+ cells was observed in the vaccinated group on weeks 2 and 6, following the prime and booster doses (respectively) (Figure 5). This increase in the B-cell population could be attributed to the elevated antibody production observed in the vaccinated cats. Interestingly, infection with *C. felis* at week 9 (0 dpi) resulted in drastic changes in the B- and T-cell populations, and an inversion in both the B-cell:T-cell ratio and the CD4+:CD8+ cells was observed following *C. felis* challenge (Figure 5). Additionally, we observed that the CTL response to the vaccine epitope c88 in the vaccinated cats was significantly correlated with CD8+ cell numbers (Appendix A), and the decreased numbers of CD8+ cells observed during infection with *C. felis* may account for the decreased cell-mediated immune response to *C. felis* challenge.

Despite the elevated and sustained humoral responses observed in the vaccinated cats, we did not observe a significant difference in the parasite load (Figure 8) between the vaccinated and sham-vaccinated groups. Concurrently, as described above, we noticed the cellular response was also not sustained in the vaccinated cats. These variations in humoral and cellular immunity have also been observed in studies on leishmaniasis, emphasizing that increased antibody levels lead to parasite persistence and weakened cellular immune responses [54]. Both cellular and humoral immune responses play an important role in infections. However, for intracellular parasites, the cellular immune response, specifically cytotoxic T-cell response, plays an important role in destroying infected cells, thereby reducing the replication and the spread of the parasites [55,56,57]. While the humoral response was high against c88, the cell-mediated response to this antigen was not sustained. It may be possible to increase both the humoral and cell-mediated responses in future studies by increasing the dose of HuAd5-c88 alone (since the dose in this study was lower than other studies in order to include HuAd5-cf76 in the vaccine administered). It is also possible that a third booster might be helpful in sustaining the cytotoxic T cell response against the *C. felis* infection. 

Although the vaccine did not prevent infection, we demonstrated that vaccination impacted the onset of clinical signs in vaccinated cats compared to the sham-vaccinated cats (Figure 7). Common clinical manifestations of cytauxzoonosis include fever, lethargy, anorexia, depression, dehydration, and icterus [2,58,59]. Experimentally infected cats typically start showing the symptoms 11–14 days after being infected with *C. felis* via tick bites [2,38,39]. Similarly, sham-vaccinated cats in the present study began showing clinical signs after 11 dpi. Clinical scores in the sham-vaccinated cats increased earlier than vaccinated cats and remained high through 17 dpi, whereas in vaccinated cats, a 1–2-day delayed onset in clinical scores was exhibited, suggesting that the immune response to vaccination may have been partially protective in reducing clinical severity. This is further supported by the decreased febrility observed in the vaccinated cats. In both vaccinated and sham-vaccinated cats, body temperatures started to increase after 8 dpi (Figure 7b). However, body temperatures in vaccinated cats remained lower compared to the sham-vaccinated cats, especially between 13–16 dpi. Studies in *C. felis*-infected cats have shown that most cats develop a fever during infection between 104–106 °F [2,38,39]. In the present study, we observed fever in sham-vaccinated cats that increased above 104 °F (40 °C), but in vaccinated cats, average body temperatures remained around 102 °F (38.9 °C) (Figure 7b). Fever is a common symptom induced by pyrogens following an infection [60]. Although febrile temperatures benefit the host by limiting the growth and the spread of the pathogen, higher body temperatures could also result in protein denaturation and changes in membrane fluid of the host, thereby worsening the disease severity [61]. However, in this study, lower febrile temperatures in vaccinated cats may have reduced clinical severity and contributed to delayed onset of clinical signs in this cohort. 

Another interesting finding of this study was the LGL response to *C. felis* infection, and importantly for immunization (Figure 6). Large granular lymphocytes are a subset of lymphocytes derived from either natural killer cells (CD3- CD16+) or cytotoxic T-cells (CD3+ CD8+ CD4-) [62,63,64]. In a healthy human or an animal, LGLs can comprise 10–15% of PBMCs [63], and different conditions such as infections can increase LGLs in these individuals [64,65,66]. As little is known about the LGL response to *C. felis* infection, we did a blind evaluation of the blood smears of each cat throughout the study period. Our results showed that infection with *C. felis* leads to a drastic increase in LGL numbers in cats. Prior to *C. felis* infection, LGLs were significantly elevated in the immunized cats in response to vaccination. Future studies will focus on understanding the role of LGLs in cytauxzoonosis and their contribution to the immune response and immunization. 

Though we demonstrated increased cellular and humoral responses to the vaccine epitopes, delayed onset of clinical signs and increased LGLs in the vaccinated cats, this vaccine was not able to prevent the cats from *C. felis* infection. The infectious stage of *C. felis* to felids is the sporozoite [14]. However, c88 and cf76, the vaccine candidates used in the present study, are known to be expressed in the schizogenous phase of *C. felis*. It could be possible that though the vaccine candidate(s) used in the study are markedly immunogenic because they are expressed in a later stage of the life cycle of *C. felis* that aggressively replicates and infects other cells, the protective mechanism may not have been able to outcompete the defensive mechanisms of *C. felis*. Future studies might be successful in focusing on employing immunogenic vaccine candidates that are expressed in the infective (sporozoite) stage of *C. felis* infection in order to initiate an expedited immune response against infection (before *C. felis* reaches the highly proliferative schizogenous phase). Studies have also shown that IL-12 is linked to long-lasting cellular immunity against intracellular parasites [67,68,69,70]. Development of vaccines that can result in persistent IL-12 could be an attractive strategy to enhance the efficacy of vaccines against *C. felis*. It should also be noted that in experimental infections such as the current study, cats are subjected to higher adult tick infestation levels in order to maximize opportunity for *C. felis* transmission, compared to lower tick burdens observed in naturally infected cats [71,72]. As a result, experimental parasite burdens of *A. americanum* and *C. felis* are likely much higher than those of cats who develop cytauxzoonosis naturally, and a clinical trial may be more suitable to assess the true efficacy of these vaccines.

## 5. Conclusions

In conclusion, we demonstrate that AdHu5 vaccine vectors can be used as a potential vaccine strategy against cytauxzoonosis. Immunization induces both cell-mediated and/or humoral immune responses to the vaccine epitopes, c88 and cf76. While the vaccine did not ultimately prevent *C. felis* infection in immunized cats, decreased pyrexia (fever) and delayed onset of clinical signs were also observed in vaccinated cats compared to the sham-vaccinated cats. Vaccination also increased the LGLs in vaccinated cats which are observed during a natural infection. Further investigations might focus on vaccines that can induce transmission-blocking antibodies, or whole parasite vaccines resulting in a broad immune response and/or multi-stage vaccines. Additionally, alternative prime-booster combinations should also be evaluated to increase the efficacy of the vaccine.

## Figures and Tables

**Figure 1 vaccines-11-00573-f001:**
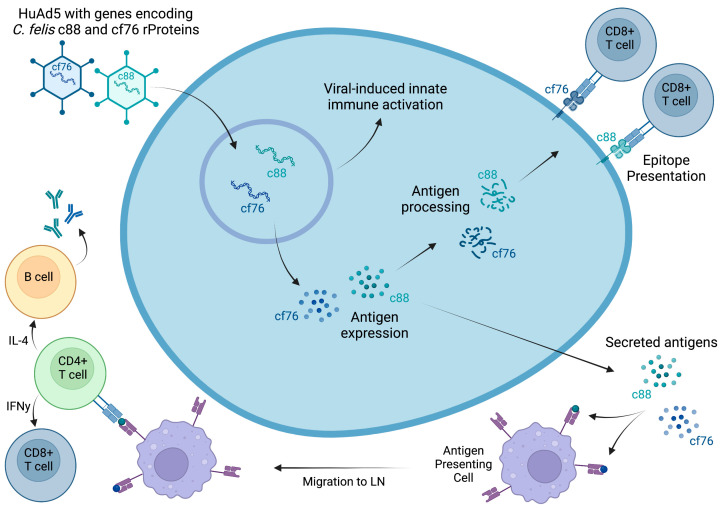
Proposed mechanism of action of the HuAd5 vector vaccine encoding putative *C. felis* antigens, c88 and cf76. Created with BioRender.com, accessed on 30 January 2023.

**Figure 2 vaccines-11-00573-f002:**
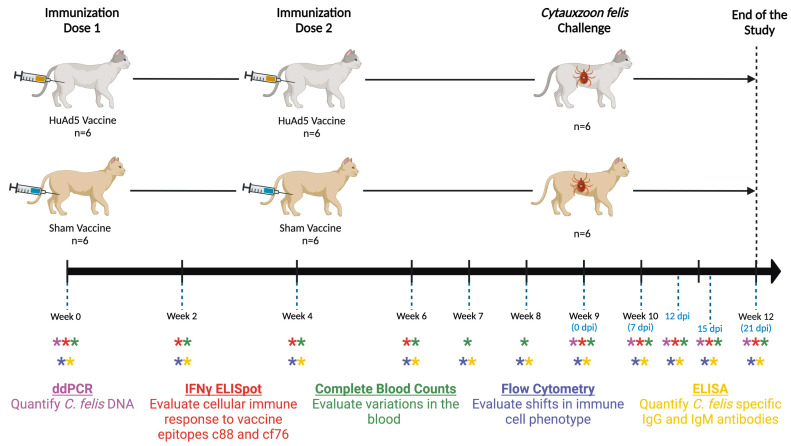
Timeline of the study indicating blood sample collection points for different assays. Different colored asterisks represent different assays carried out at each sampling point which are listed in corresponding colors at the bottom of the figure. Created with BioRender.com, accessed on 3 February 2023.

**Figure 3 vaccines-11-00573-f003:**
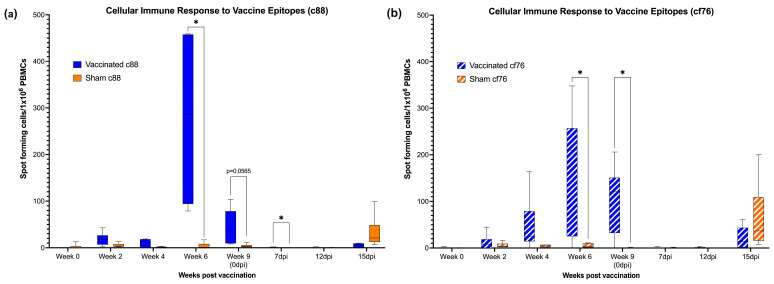
Cellular immune response to vaccine epitopes following the vaccination in vaccinated and sham-vaccinated groups: (**a**) c88; (**b**) cf76. Vaccination induced a strong cellular immune response in the vaccinated cats which was peaked after the booster dose. However, the cellular response decreased over time and was not sustained during the course of infection. * *p* < 0.05.

**Figure 4 vaccines-11-00573-f004:**
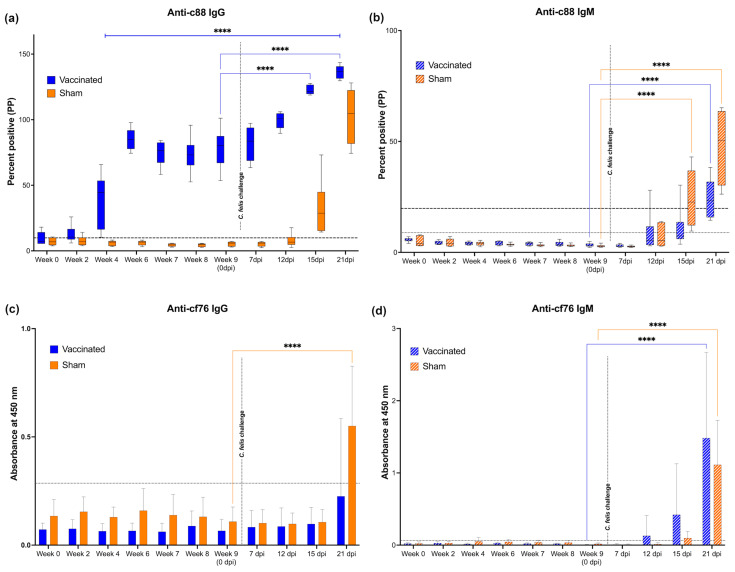
*C. felis*-specific anti-c88 and anti-cf76 antibody levels in cats: (**a**) anti-c88 IgG; (**b**) anti-c88 IgM; (**c**) anti-cf76 IgG (**d**) anti-cf76 IgM. A pronounced anti-c88 IgG response was observed in vaccinated cats at week 4 post-vaccination which was sustained throughout the study, including the infection challenge. A notable anti-cf76 IgG response was not observed in response to vaccination cf76. Dashed horizontal line in (**a**) indicates the strong positive cutoff, as reported in [42]. Upper dashed horizontal line in (**b**) indicates strong positive cutoff, while the lower dashed horizontal line indicates weak positive cutoff, as reported in [42]. Dashed horizontal lines in (**c**,**d**) indicate positive cut-off values established as previously described [43]. **** *p* < 0.0001.

**Figure 5 vaccines-11-00573-f005:**
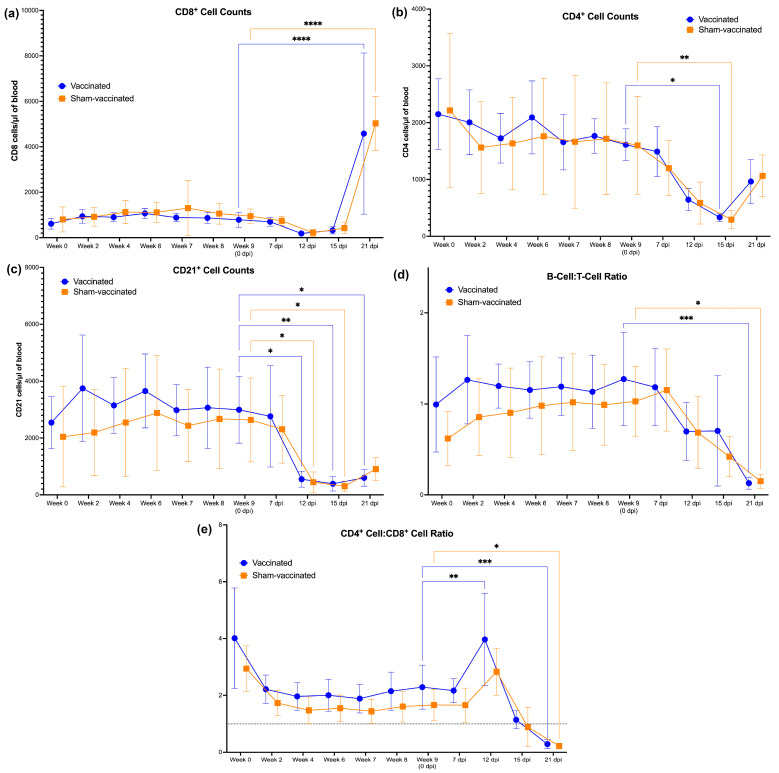
The number of B and T cells per one microliter of whole blood in the cats during the course of the study: (**a**) CD8+ cells; (**b**) CD4+ cells; (**c**) CD21+ cells; (**d**) B- to T-cell ratio; (**e**) CD4+ to CD8+ cell ratio. Vaccination did not significantly change the lymphocyte composition in vaccinated cats. However, infection with *C. felis* resulted in increased CD8+ cells (at 21 dpi) and decreased CD4+ and CD21+ cells in both groups of cats. **** *p* < 0.0001; *** *p* < 0.001; ** *p* < 0.01; * *p* < 0.05.

**Figure 6 vaccines-11-00573-f006:**
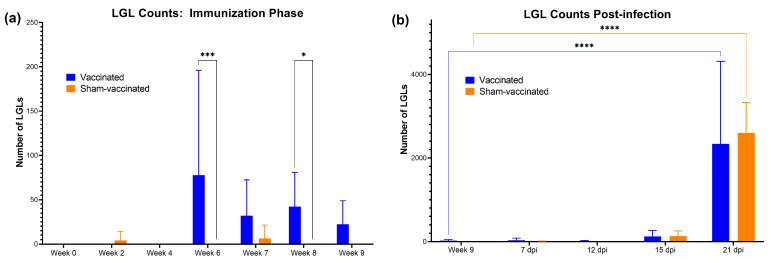
Total number of large granular lymphocytes (LGLs) per μL of blood: (**a**) pre-infection; (**b**) post infection. Vaccinated cats had significantly higher LGLs during the immunization phase. *C. felis* infection significantly increased LGLs in both vaccinated and sham-vaccinated groups. **** *p* < 0.0001; *** *p* < 0.001; * *p* < 0.05.

**Figure 7 vaccines-11-00573-f007:**
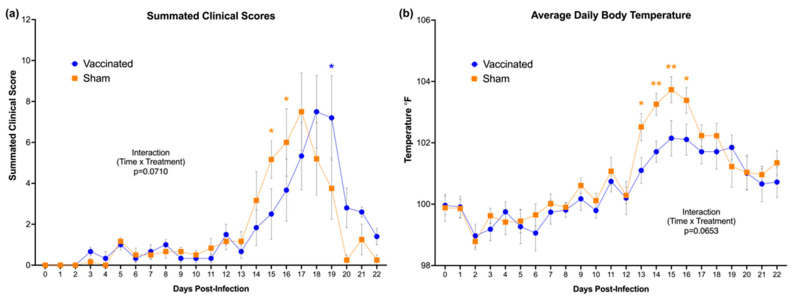
(**a**) Average clinical scores of the cats following *C. felis* challenge. Following tick infestation, cats were evaluated daily for clinical parameters listed in Table 1 and a score was assigned for that day. Vaccinated cats showed significantly lower clinical scores and a delayed onset of cytauxzoonosis. (**b**) Average daily body temperatures of the cats during *C. felis* challenge. Vaccinated cats exhibited lower average body temperatures during *C. felis* challenge when compared to sham-vaccinated cats. ** *p* < 0.01; * *p* < 0.05.

**Figure 8 vaccines-11-00573-f008:**
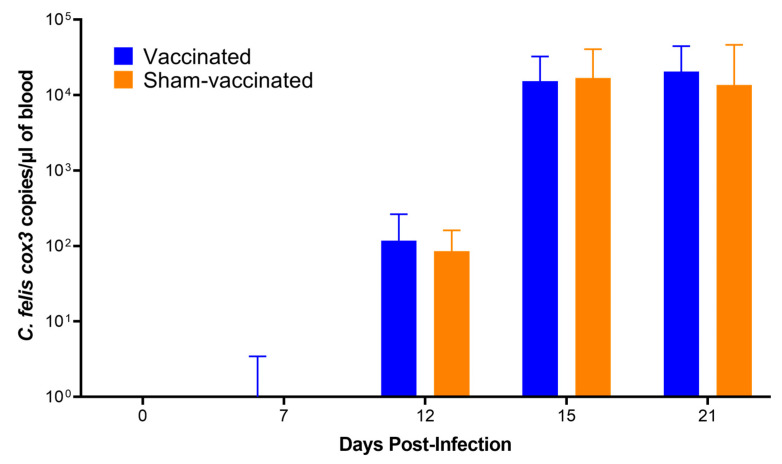
Parasite loads of the cats following the *C. felis* challenge. All cats became infected with *C. felis* during challenge and there was no significant difference between *C. felis* cox3 gene expression at any point during this study.

**Figure 9 vaccines-11-00573-f009:**
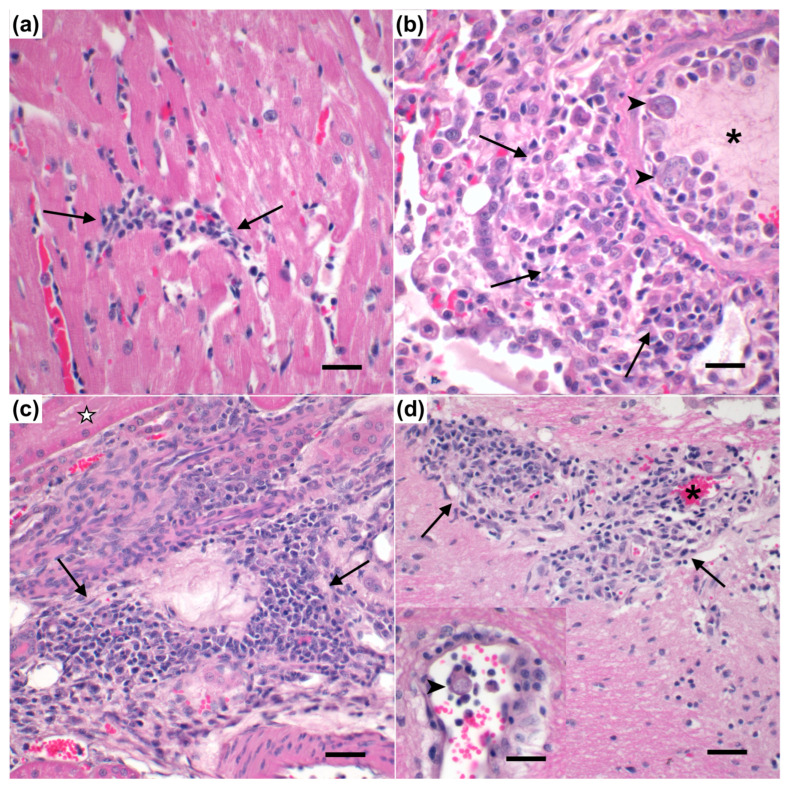
Histopathology of *C. felis* infection in sham-vaccinated cats. (**a**) Sham-vaccinated cats exhibited slightly increased mononuclear inflammation (arrows) in the interstitium of the heart. (**b**) The lungs of sham-vaccinated cats exhibited pronounced mononuclear inflammation (arrows) and occlusion of vessels (asterisk) with intravascular schizonts (arrowheads) containing 1–3 µm merozoites. (**c**) Renal tubules (☆) were effaced by mononuclear inflammatory infiltrates (arrows). (**d**) Cerebral vessels (asterisk) were disrupted by mononuclear infiltrates (arrows). Cerebral vessels occasionally contain intravascular schizonts (arrowhead). Magnification: (**a**,**b**, inset **d**) 60×, scale bar = 12 μm; (**c**,**d**) 40×, scale bar = 25 μm.

**Table 1 vaccines-11-00573-t001:** Clinical scoring system for acute cytauxzoonosis. The system was developed to evaluate the daily clinical scores of the cats following tick infestation to assess differences in the clinical signs between the vaccinated and sham-vaccinated cats. Each cat was examined daily in the morning and scores were assigned from 0–3 for each clinical parameter. Evaluators were blinded to the immunization status of the cats to ensure unbiased assessment and to initiate treatments as needed. * Disturbed: observer in the room but kennel unopened. ** Stimulated: kennel open.

Clinical Parameter	0	1	2	3
Rectal Temperature	100.5 to 102.5 °F	102.5 to 103.5 °F	103.6 to 105 °F	>105.1 °F
MM CRT	Pink, moist <2 s	Pale pink, tacky 2–3 s	Pale, tacky to dry 2–3 s	Pale or bright red >3 s
Activity	Normal	Mild reduction when disturbed * (mild lethargy)	Moderate reduction when disturbed * (moderate lethargy)	Little to no activity Disturbed * and reduced activity stimulated **
Appetite	Normal	Reduced interest in food; 1 d not eating	Markedly reduced interest in food; 2 d not eating	Anorexia; 3 d not eating
Respiratory Effort	Normal resting respiratory rate and normal effort	Mild tachypnea (>35 breaths per min); no overt increase in effort otherwise	Moderate tachypnea (>40 breaths per min); moderate increase in effort	Marked tachypnea (>45 breaths per min); marked effort or dyspnea
Icterus	None	Mild scleral icterus; none in skin or MM	Subtle icterus in sclera, skin, and MM	Obvious icterus in sclera, skin, and MM
Dehydration	Euhydrated	~5%—Mild, semi-dry oral mucus membranes, normal eye	~6 to 8%—moderate degree of decreased skin turgor; dry oral mucous membranes	>8%—marked degree of decreased skin turgor, dry mucous membranes, weak and rapid pulse, slow CRT, notable mental depression
Pain	≤2.0	2.0 to 2.25	2.5 to 2.75	≥3.0

## Data Availability

All data will be published in a public repository upon acceptance of publication.

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
