# Peer review of "A Novel Vaccine Strategy to Prevent Cytauxzoonosis in Domestic Cats"

_vaccines, 2023, doi:10.3390/vaccines11030573_

Round 1
Reviewer 1 Report
Very good work,.
Comments are included in the manuscript (pdf)

Author Response
Manuscript ID: vaccines-2230484
Response to reviewers
Thank you so much for giving us an opportunity to submit a revised draft of the manuscript titled “A Novel Vaccine Strategy to Prevent Cytauxzoonosis in Domestic Cats” for publication in Vaccines. We appreciate the time and effort that your editorial team and the reviewers dedicated for providing feedback on our manuscript. We have addressed and incorporated the suggestions made by the reviewers. Please see below for a point-by-point response to the reviewers’ comments and concerns. All page and line numbers refer to the revised manuscript file with tracked changes.
Reviewer #1’s Comments
- VERY IMPORTANT IDEA TO REMARK, IT REDUCE THE DISEASE IMPACT BUT IT DOES NOT PREVENT INFECTION
Author response: We sincerely appreciate your valuable comments and the thorough evaluation.
- IN MY OPINNION RESULTS OF THE STUDY SHOULD NOT BE INCLDUED IN THE INTRODUCTION, PERHAPS IT COULD BE TRANSFORME IN OBJECTIVES OF THE CURRENT SUTDY
Author response: The text has been modified accordingly
- IT SHOULD BE EXPLAINED PREVIOUSLY IN POINTS EXPLAINING THE DESIGN OF THE STUDY, IT IS NOTAN ANALYSIS
Author response: Thank you for your comment. We believe that mentioning the cats were randomly grouped in this section is relevant to the statistical analyses done in this study, and that makes it easier for the reader if it is mentioned in this section rather than in the study design section. Therefore, we did not remove or change that sentence (line 292) in the revised manuscript.
- I UNDESTAND THAT 6 ANIMALS PER GROUP IS A SMALL SAMPLE SIZE, I SUGGEST TO COMMENT IT BECAUSE YOU SAID: WAS SIGNIFICANTLY INCREASED... DOES IT MEANS ACCEPTABLE STATISTICAL SIGNIFIICANCE?
Author response: Yes, for each time we reported values to be significantly increased or decreased, we compared the values of the vaccinated group to the sham-vaccinated group by statistical methods and used the word “significant” to denote acceptable statistical significance. This is addressed in section 2.12 via the text “p-values < 0.05 were considered statistically significant.”
- ATTENTION, SOMETIMES IN ITALIC SOMETIMES NOT, UNIFORMITY OF USE.
Author response: Thank you for pointing this out! We thoroughly screened through the manuscript and italicized the scientific names accordingly and made sure it is consistent throughout the manuscript.
- SAME OPINNION THAN PREVIOUSLY ABOUT SIGNIFICANTLY
Author response: Please see the response to 4.
- THIS MEANS PART OF THE ANIMALS DID NOT COMPLETED THE STUDY WHICH COULD HAD AFFECTED TO THE ANALYSIS AND THE LEVEL OF SIGNIFICANCE, DON´T YOU AGREE?
Author response: In this study, we were able to collect samples from all the animals up to 15 dpi. The only time point where we had only 4 sham vaccinated cats and 5 vaccinated was at 21 dpi. We used repeated measures ANOVA with mixed effects analysis for all the statistical analyses to account for these missing values. Any reported statistical significance is reported with this data taken into account. While missing values at 21 dpi may have slightly decreased our ability to detect statistical significance, we believe this was the most appropriate and unbiased way to analyze the data.
- I LIKE THIS SENTENCE, PERHAPS IT COULD BE INTERESTING THAT YOU EXPLAIN BECAUSE THIS "SPECULATION" AND NOT "DEMONSTRATION" (LATTER, IN CONCLUSIONS YOU BEGIN USING DEMONSTRATED..
Author response: In line 591, we speculate the overall decreased clinical manifestations in vaccinated cats, especially 15-16 dpi, could be attributed to the overall low body temperatures in the vaccinated group during the infection. In line 608, we are mentioning that this study demonstrated a delayed onset of clinical signs in the vaccinated cats.
Reviewer 2 Report
The paper on a potential vaccine against cytauxzoonosis in cats is well written and experimentally sound. A few minor changes are suggested:
line 26, delete 'lowered overall clinical scores,'. The data did not show this.
lines 28-29, end the last sentence at ... against cytauxzoonosis. The rest of the sentence is tangent to the main point of the paper and was sufficiently discussed in the Discussion.
line 339, I suspect the ... in anti-c88 IgM ... should by anti-cf76
in section 3.3 some mention that CD21+ are B-cells would be helpful for many readers
line 499 not really sure that the results for c88 were fully successful. cell-mediated response was quite transient
line 571, the data did not show that clinical signs other than fever were lowered
line 577, not sure the delayed onset was significant. perhaps: ... cats exhibited a 1-2 day delayed onset ...
lines 590-593, perhaps delete the last sentence. The data doesn't really support the spectulation and not sure if any speculation is needed at that point
Other minor editorial suggestions:
line 17, change that to and
line 22, delete against C. felis
line 24, delete significant
line 25, delete ultimately, delete with C. felis
line 50, insert and between feeding/interupting
line 51, change and/or to or (or is inclusive)
line 346, delete Interestingly,
line 535, change showcase to exhibit
line 564, delete very
Author Response
Manuscript ID: vaccines-2230484
Response to reviewers
Thank you so much for giving us an opportunity to submit a revised draft of the manuscript titled “A Novel Vaccine Strategy to Prevent Cytauxzoonosis in Domestic Cats” for publication in Vaccines. We appreciate the time and effort that your editorial team and the reviewers dedicated for providing feedback on our manuscript. We have addressed and incorporated the suggestions made by the reviewers. Please see below for a point-by-point response to the reviewers’ comments and concerns. All page and line numbers refer to the revised manuscript file with tracked changes.
Reviewer #2’s Comments
The paper on a potential vaccine against cytauxzoonosis in cats is well written and experimentally sound. A few minor changes are suggested:
Author response: Thank you so much for taking time to review and providing positive comments to improve our manuscript.
- Line 26: delete 'lowered overall clinical scores,'. The data did not show this.
Author response: Thank you for your comment. The text has been modified accordingly.
- Line 28-29: end the last sentence at ... against cytauxzoonosis. The rest of the sentence is tangent to the main point of the paper and was sufficiently discussed in the Discussion.
Author response: This change has been made in the revised manuscript.
- Line 339: I suspect the ... in anti-c88 IgM ... should by anti-cf76
Author response: In this section we are comparing the IgG to IgM levels for c88.
- in section 3.3 some mention that CD21+ are B-cells would be helpful for many readers
Author response: Thank you for pointing this out. We have added this to the manuscript.
- Line 499: not really sure that the results for c88 were fully successful. cell-mediated response was quite transient
Author response: The results of our study do show that the IgG antibody response to the c88 vaccine epitope was increased following the first vaccine dose which further increased and sustained following the booster dose (Figure 4a). This was not observed for the vaccine epitope cf76. Vaccination also induced cellular immune response against c88 (Figure 3a), though it didn’t sustain and increased as expected following the infection with C. felis. We have discussed these in the manuscript (lines 531-537).
- Line 571: the data did not show that clinical signs other than fever were lowered
Author response: This has been modified accordingly.
- Line 577: not sure the delayed onset was significant. perhaps: ... cats exhibited a 1-2 day delayed onset ...
Author response: This has been modified accordingly.
- Line 590-593: perhaps delete the last sentence. The data doesn't really support the speculation and not sure if any speculation is needed at that point
Author response: Thank you for this comment. The text has been modified to be reflective of the data (delayed onset of clinical signs and reduced clinical severity as evidenced by a decreased need for ancillary therapy in vaccinated cats with slightly decreased mortality).
Other minor editorial suggestions:
- line 17, change that to and
Author response: The text has been modified to improve readability.
- line 22, delete against C. felis
Author response: Deleted in the revised manuscript.
- line 24, delete significant
Author response: Thank you for this comment, but we respectfully argue that the humoral and cellular response in the vaccinated cats were significantly higher in the vaccinated cats compared to the sham-vaccinated cats.
- line 25, delete ultimately, delete with C. felis
Author response: We chose to leave the word “ultimately” in the revised manuscript as one of our aims was to test if this vaccine strategy could ultimately prevent infection.
- line 50, insert and between feeding/interrupting
Author response: This is added to the line 50.
- line 51, change and/or to or (or is inclusive)
Author response: Corrected as suggested.
- line 346, delete Interestingly
Author response: The text has been modified accordingly.
- line 535, change showcase to exhibit
Author response: The change was made in the revised manuscript.
- line 564, delete very
Author response: Deleted as suggested.